# Working Memory Training: Assessing the Efficiency of Mnemonic Strategies

**DOI:** 10.3390/e22050577

**Published:** 2020-05-20

**Authors:** Serena Di Santo, Vanni De Luca, Alessio Isaja, Sara Andreetta

**Affiliations:** 1Cognitive Neuroscience, Scuola Internazionale Superiore di Studi Avanzati, I-34136 Trieste, Italy; sd3362@columbia.edu (S.D.S.); isaja@sissa.it (A.I.); 2Scuola Peripatetica d’Arte Mnemonica (S.P.A.M.), 10125 Turin, Italy; info@vannideluca.it

**Keywords:** working memory, mnemonic strategies, complex span, memory training

## Abstract

Recently, there has been increasing interest in techniques for enhancing working memory (WM), casting a new light on the classical picture of a rigid system. One reason is that WM performance has been associated with intelligence and reasoning, while its impairment showed correlations with cognitive deficits, hence the possibility of training it is highly appealing. However, results on WM changes following training are controversial, leaving it unclear whether it can really be potentiated. This study aims at assessing changes in WM performance by comparing it with and without training by a professional mnemonist. Two groups, experimental and control, participated in the study, organized in two phases. In the morning, both groups were familiarized with stimuli through an N-back task, and then attended a 2-hour lecture. For the experimental group, the lecture, given by the mnemonist, introduced memory encoding techniques; for the control group, it was a standard academic lecture about memory systems. In the afternoon, both groups were administered five tests, in which they had to remember the position of 16 items, when asked in random order. The results show much better performance in trained subjects, indicating the need to consider such possibility of enhancement, alongside general information-theoretic constraints, when theorizing about WM span.

## 1. Introduction

Since the publication of the seminal paper by Baddeley and Hitch in 1974 [1], working memory (WM) has become a central notion, both in experimental paradigms and in theoretical models. The authors hypothesized a system, distinct from short-term memory, in which a control mechanism could temporarily hold information in order to process it. Their hypothesis rose from the observation that, in test settings, when the information to be remembered exceeds the capability of a temporary storage, then another mechanism is engaged to provide support. This mechanism was later called the central executive, which acts as a controller for coordinating the activity of two other, modality-specific stores: a visuo-spatial pad for visual images and a phonological buffer for linguistic material. Given this articulation, it is tempting to speculate that, while the pad and the buffer may have adapted to the material they usually process, the central, amodal mechanism may be subject only to neural and information-theoretic constraints.

Many studies have then adopted this view and tried to disentangle the dynamics of WM with the result, however, of a definition which is still rather vague today. Most scholars [1,2,3] have agreed on describing it as a system capable of computing operations while keeping information in a buffer in order to process it. Indeed, being able to execute operations is what distinguishes it from a classical short-term storage, taken to be a system with which it is possible only to rehearse information ‘as is’, over a brief interval of time. This less flexible system is generally considered to hold discrete items, or pre-assembled chunks of items, and its short-term span is defined as the longest list of items a person can retain in a task requiring immediate recall. 

The capacity of the two systems is therefore measured with different tasks. Usually, tasks that are employed to measure the short-term memory span are referred to as simple span tasks [4]. In general, they consist in retrieving a list of items previously memorized; they are performance-adapted, with difficulty increasing proportionally to the person’s memory span [5]. In contrast, since WM refers not only to a storing process, but also to an operating ability, tasks which measure WM capacity should require performing some operation. Typically, then, WM is assessed through complex span tasks where, in addition to the retrieval of items, participants are also asked to execute some computation on the items. It can be mathematical operations, decisions, or judgements. However measured, complex span is found to be rather limited, reinforcing the view [6] that it may be subject to domain-independent constraints, related to physiological restrictions.

Notably, the literature indicates that performance in the complex span tasks is often associated with several cognitive abilities. As a consequence, WM capacity is associated with such abilities, with the result of being considered an important predictor in many cognitive domains. For instance, it has been correlated with multi-tasking [7], executive functions [8], language abilities [9,10] and with the capacity to reason with novel information [11,12]. Conversely, a lower WM capacity has been associated with cognitive disorders, in particular with attention-deficit hyperactivity disorder [13] and with learning difficulties [14].

Albeit not addressing the complicated issue of causality between WM capacity and cognitive abilities, these findings lead to the question of whether WM capacity can be trained; if so, in fact, it might imply that many cognitive abilities could benefit from such training. Quantitative evaluations, however, are controversial, and many potentially contributing factors should be considered. 

Several different tasks and methodologies have been used during the past decades to measure the effects of training on WM, and, often, it is this heterogeneity which causes difficulty in interpreting the results. A first point which needs to be considered, for instance, is whether training improves WM capacity or efficiency. According to von Bastian and Oberauer [15], in the first case the training would enable the subject to hold more items in WM than before, whereas an efficiency benefit would still use the capacity available but would rely, for instance, on additional strategies to retrieve more complex items (e.g., chunking them into larger units). Therefore, training WM capacity would be independent of specific tasks, while training efficiency could be expected to be material- or process-specific.

In their review, von Bastian and Oberauer noted the large variability in several aspects of WM training in the different studies, which should be taken into account when evaluating its effects: (a) the intensity and duration of training, which they observe to range from three to one hundred sessions, and from ten to forty-five minutes for every session; (b) the adjustment of task difficulty, of unclear relevance, as in some cases results show no differences in benefits of training between an adaptive and a non-adaptive procedure; (c) control training: even this aspect is controversial, whether training for the control group should include the same tasks, but at a lower level of engagement or difficulty, or else it should be based on other tasks, so as to affect WM capacity as little as possible. In both cases, in fact, motivation is an important factor to consider. This aspect was also emphasized in a comprehensive review by Shipstead and colleagues [5]: the authors mentioned the lack of a control group in many WM training studies. Even when a control group is involved, this may bring in unintended effects: sometimes, for instance, control subjects comprise a so-called “no-contact” group, because they just participate in pre- and post-training test sessions, without a major motivational involvement in the experiment. This should be expected to influence results, as also noticed by von Bastian and Oberauer [15]. In order to elicit a better engagement of the control group then, it would be beneficial to administer it a shallow version of the training. In this case then, a possible placebo effect also needs to be considered, and perhaps controlled by providing the same information about training to both groups. 

On this line, von Bastian and Oberauer also pointed out that it is necessary to take into account whether a potential improvement in WM performance depends on WM training or is a general effect of training per se. To address this issue, they suggest that training should focus specifically on WM tasks. On the other hand, WM training should not teach strategies valid only for one task, but it should be applicable cross-tasks. If aimed at a single task, its effects might not be transferable to other paradigms, e.g., involving words, digits or images. In this respect, some studies have shown that some methodologies have an effect on a subtype of WM and not on another one. For example, experiments with order recall tasks indicate that the order of presentation is influential for verbal, but not for visuospatial memory [16]. Therefore, attention to the exact type of WM is also important. 

Furthermore, in the context of an enhanced WM as a beneficial effect for other cognitive domains, Shipstead and colleagues claimed that there is a difference between a near transfer and a far transfer. Near transfer is taken to mean an improved performance in tasks used in the experiment, whereas far transfer indicates an additional benefit that could be manifest later in related abilities, beyond those assessed with the experimental tests [5].

To summarize these controversies, the road to WM training is far from being a clear path. Our aim is to quantitatively assess the efficiency of some classic mnemonic strategies applied to WM tasks. We think that our study can help in building a quantitative framework in the field, in that we carefully chose the logic of the experiment in order to have a grounding that is as solid as possible:We carefully chose WM tasks which require the participants to perform an operation (as by definition of WM tasks), but explicitly looking for an operation that requires as little computational power as possible. It is in fact hard to assess how many resources the participant needs to employ in order to perform a simple computation, as required for a complex span task. We then decided for a perturbative approach, where the requirement of executive control processing is devoted to the random retrieval order: the participant is asked to be able to navigate across the formed memories.The training of the experimental group consists of “classic” mnemonic strategies. These strategies did not wait, of course, for the formulation of the concept of WM: they are much more ancient. Among their most prominent promoters, in fact, we find the Roman orator Cicero (106 b.C.–43 b.C.), we find them mentioned even in older Greek treatises and, within the more recent literature, they represent the main topic of “Moonwalking with Einstein” [17], a book describing Joshua Foer’s journey from being a journalist to winning the international memory competition, confirming that their efficacy is well demonstrated in several contexts. These techniques include, in particular, the “method of loci” and the “memory palace”. Even though the transmission of these strategies throughout the centuries confirms per se their value, to the best of our knowledge they have never been tested in a lab setting with naïve participants. One of us, VdL, is a professional mnemonist who relies on these strategies for his own on-stage performance, and he participated in this study by offering 2 h of mnemonic strategy training to our experimental subject group.The training is kept short and the possibility to practice the freshly learnt strategy is very limited. Obviously, professional mnemonists dedicate a lot of time to the practice of these techniques. This is confirmed by Foer’s description of his experience [17] and by VdL as well. In their case, in fact, the outstanding performances achieved are to be imputed both to the use of successful strategies and to the amount of practice that they devote to improving a specific skill set. In our setup, the training being limited to one short session, possible differences between the control and the experimental group are likely to be due to the use of a better strategy.The WM span is usually considered largely domain-independent [6]. This was an important factor in our design: in our experiment, we included different types of material, in order to check whether the same training can benefit different tasks or, in other words, whether the newly learnt strategy could be generalized to be used on different items.

Moreover, the strategies offered to the experimental group require a recoding process: different stimuli are transformed into a similar pattern: with the “method of loci”, numbers and words are converted into images, and put in different locations (for a detailed description see par. 2.4). To recall items then, participants go along a trajectory, throughout the path of the locations they used. This may facilitate putting together chunks of information: in the same location we can put different items. Indeed, with a single image or location we can memorize shorter or longer words, and digits as well, and it is conceivable to use it to hold in memory what would have otherwise been several items. The role of chunking was discussed also by Miller in his seminal paper on the “magical” number seven, when interpreted as the span for short-term memory [18]. It implies that each chunk can include a variable number of bits of information; therefore, if what is limited is the number of chunks, it seems hopeless to cling to the notion of an information-theoretic bound on short-term memory retention. When one manages to construct larger chunks, more bits will be retained. Recoding, whether in a verbal narrative when describing an event, as mentioned by Miller, or in a set of location-image associations as envisioned by the methods of loci, plays then a critical role in potentially stretching the limits of our memory capacities. 

Two groups participated in the experiment: an experimental group, who attended the two-hour seminar in which these strategies were described and applied, and a control group, who attended instead a two-hour seminar on memory systems in the brain, by another one of us, SA. The goal was to assess whether the group trained by VdL benefited from the mnemonic training in comparison with the control group.

## 2. Materials and Methods 

### 2.1. Participants

Forty-three subjects participated in the experiment, 22 in the experimental group (f = 16; average age = 27.9, sd = 4.9) and 21 in the control group (f = 18; average age = 27.01, sd = 4.9). They were recruited through the SISSA recruiting platform, and most were students at the local University of Trieste (Italy). 

None of the participants had a previous history of psychiatric or neurological illness, learning disabilities, or hearing or visual loss. 

They were asked to participate in a study on memory, which would involve attending a 2-hour lecture in the morning and taking some memory tasks lasting at most 30 min in the afternoon, without specifying that in the lecture they would be trained (or not) to use mnemonic strategies. In this way, we tried to keep the motivation high also in the control group.

Their participation was compensated with 30 euros.

### 2.2. Experimental Procedure

The experiment was conducted on two different days, one for the experimental and one for the control group. Within every group, participants were in the same computer room, where each one was assigned a workstation, so that they could be tested separately but simultaneously.

Participants were engaged for slightly over two hours in the morning and then, after a three-hour break, they continued in the afternoon with the second part of the experiment. 

The morning session included a brief N-back task followed by a seminar, which was different for the control and the experimental group. The afternoon session was comprised of 5 memory tasks, where participants could use any strategy they wished, and those in the experimental group had the opportunity to apply the techniques they had been exposed to in their morning seminar.

### 2.3. N-Back

In this task, participants saw a sequence of images, while also hearing the noun of the item. For every item, they were asked whether it was the same as three items before. The task was self-paced, and the response was given by pressing a key.

The set of items was assembled by first selecting words from the Subtitle-based Word Frequency Estimates for Italian corpus (SubTLex) [19]. We chose a pool of 48 content words that could be easily represented by a simple image and had usage frequencies in an intermediate range (from 1.05 per million for ‘bruco’ to 69.6 per million for ‘chiave’). We then selected corresponding images from the online archive [20], with permission (see Appendix B).

### 2.4. Seminar

Participants attended to the seminar right after the N-back task, in the same room. The topic of the seminar differed between the two groups.

For the experimental group, VdL explained several techniques useful to memorize information. In particular, he focused on the general “method of loci” and on the specific “memory palace” technique. The idea, he explained, is to assign a location to every object: in the “memory palace”, a position in one room, typically in a corner. Therefore, every room can contain four objects, one in every corner, and is located inside a building, where the subject can mentally explore as many rooms as are needed to store all the items. Ideally, the rooms and buildings chosen are very familiar. 

The technique can be used to help memorize any sequence of items, presented verbally or visually. For digits, a common strategy is to supplement it with a visual trick, whereby digits are associated with vivid images of the same shape, e.g., a “0” with a cookie, or a “2” with a swan. 

VdL alternated explanations with practical exercises, in which he (cheerfully) challenged participants to apply the technique and recall information they had memorized together. This was all done as a group, in 2 h.

The control group also attended to a 2-hour seminar, aiming to induce a comparable level of sustained attention, participation and fatigue. It was also about memory, but not about mnemonic strategies: SA described memory systems and narrated anecdotes about clinical cases, prodigious memories and flashbulb memories. These topics were carefully chosen in order to encourage an active participation with questions and curiosities by the participants, and to balance (as much as possible) the level of motivation of the control group with the experimental group. (The gender of the person who imparted the seminar to the control group and the experimental group was not the same, potentially introducing a bias. Here we are working under the assumption that this bias does not produce major effects).

After the seminar, participants had a three-hour break.

### 2.5. Memory Tests

After the break, participants got back to their workstation, where they were administered 5 memory tests, in which they could of course use any strategy they wished, including applying the technique discussed by VdL with the experimental group. The material included words, digits and images from the same pool described in Section 2.3. The 5 tests are described in the next paragraphs in the order in which they were administered, which was therefore the same for all participants. Within each test, however, the order of individual items was randomized separately for each participant.


**Test 1: Word sequence**


Sixteen words were presented sequentially in auditory form, with individual headphones. Every audio stimulus was associated with a number, which was shown on the screen on the subject’s workstation. The presentation thus followed a numerical order. Participants had to memorize at their own pace the word they heard, together with its ordinal position. 

After this learning phase, a customarily coded software asked the participants to retrieve the words they memorized, but in random order. A random number from 1 to 16 (without repetitions) appeared at each of 16 trials, and they could choose the correct word from a visual pool of 48 written words, presented in a 6 × 8 grid at the bottom of the screen. They could pick the correct item by clicking on it (see Figure 1). After they clicked, the chosen item appeared next to the number. No feedback was given.


**Test 2: Image sequence**


Sixteen images were shown sequentially at the center of the screen, covering an area of ca. 5 × 5 cm. Again, every image appeared together with a number, and participants had to memorize at their own pace the image they saw, together with its associated number. They were then asked to retrieve the images they memorized, again in random order. A number appeared and they could choose the correct image from a pool of 48 images, presented again in a 6 × 8 grid, but on a separate screen, accessed through an arrow (see Figure 2). Unlike the previous test, then, the items were stored and retrieved in the same modality.


**Test 3: Digits**


Sixteen digits (1 to 9) were randomly assigned to a 4 × 4 grid on the screen, obviously with repetitions. Participants had to memorize at their own pace the digit at every position in the grid. Once they were ready, the digits disappeared together. Then, a random cell lit up and they had to insert the correct digit into it. The list of digits was available on the same screen in a vertical line to the right, so they could directly click on the one they chose (see Figure 3). As with all other tests, no feedback was given.


**Test 4: Word grid**


Sixteen words were presented on the screen, again in a 4 × 4 grid, which participants had to memorize at their own pace. Once they were ready, the words disappeared, and a cell would light up at a random position. The subject had to pick the correct word for that position from the pool of 48 words, available in a 6 × 8 grid at the bottom of the screen, by clicking on it. After they clicked, the chosen item appeared in the lightened cell, and a new cell would light up (see Figure 4). Again, no feedback was given.


**Test 5: Image grid**


For this test, 16 images were presented on the screen in a similar 4 × 4 grid, and the procedure was the same as for test 4, except that the pool of 48 images was available on a separate screen, accessed by clicking on an arrow (see Figure 5)).

## 3. Results

Raw data are available in Appendix A.

### 3.1. N-Back

Figure 6 shows the results of the N-back test. Both groups are quite accurate in their answers, with an almost equal performance: 89% ± 5% for the controls and 90% ± 6% for the experimental group (mean ± SD; Mann–Whitney U (MW-U) test *p* = 0.5). The color code is consistent throughout the paper, with a yellow color (or warm palette) for the control group and blue (cold palette) for the experimental group. 

This comparable baseline performance provides a good starting point in order to interpret any differential result in the afternoon session, which we would like to ascribe to the effect of training.

### 3.2. Afternoon Tests

For each test, we report the fraction of correct answers given by participants in the two groups. Figure 7 shows the performance of each individual (small yellow dots for control subjects, big blue dots for experimental subjects) as well as the average performance (light yellow solid line for the control group, light blue solid line for the experimental group). The shaded regions indicate the region determined by mean ± SEM. The differences in the performance of the two groups are significant for all stimuli conditions (MW-U test, WS: *p* < 10^−4^; IS: *p* < 10^−4^; GD: *p* = 0.004; GW: *p* = 0.006; GI: *p* = 10^−4^).

It is clear from this figure that participants in the experimental group showed a beneficial effect of being exposed to the mnemonic strategies in the seminar: indeed, for every test, we find many subjects at ceiling in the experimental group: 45% for word sequences, 68% for image sequences, 64% for the digits and word grids, and 68% again for image grids, respectively, with 62% overall. In contrast, fewer control subjects were at ceiling, in each test: none for word sequences, 14% for image sequences, and 28% for the three grid tests, for an average of 20% overall. Note that almost half of the experimental group is at ceiling in the word sequence test, and no one in the control group. In general, this first test is seen to be the most challenging for both groups, and the one with the largest difference between them.

Figure 8 shows the group-averaged fraction of correct answers, in each test, when the items are classed in quadruplets depending on their presentation position/order. Each test was in fact designed with a total of 16 items, also to facilitate this analysis. For the two sequence tests (left panel), the four quadruplets were presented successively, while for the three grid tests (right panel), they were presented simultaneously, in the top and bottom, left and right quadrants of the grid.

We expect primacy and recency effects, possibly suppressed by close-to-ceiling performance in the experimental group. Moreover, we expect to see primacy and recency in the sequences, but less so in the grids, where an ordering is not univocal and different participants could be intuitively sectioning the grid in different ways (e.g., by columns, by rows, center versus periphery…), thus the effect would be washed out in the average. 

The control group had a significantly better performance on the first and last quadruplet of sequentially presented stimuli (MW-U test, p(1&4 vs 2&3, SW&SI) = 0.0016), indicating robust primacy and recency effects. When analyzed in more detail, the effect was salient with words p(1&4 vs 2&3, SW) = 0.003, but only marginal with images, p(1&4 vs 2&3, SI) = 0.058, and this was largely due to differences in performance indicating primacy, in particular between the first and third quadruplets (p(1vs2, SW) = 0.009 and p(1vs3, SW) = 0.003 with words, while p(1vs2, SI) = n.s. and p(1vs3, SI) = 0.01 with images; all other quadruplet comparisons were not significant). 

For grids, the primacy and recency effects in the control group are not prominent (MW-U test, p(1&4 vs 2&3, GD&GW&GI) = 0.48). Also, when considering them separately, in the different experimental conditions, significancy levels are not reached in any of the considered experiments and there is only a weak trend for the digits (MW-U test, p(1&4 vs 2&3, GD) = 0.14, p(1&4 vs 2&3, GW) = 0.39, p(1&4 vs 2&3, GI) = 0.62).

Such primacy and recency effects were suppressed, presumably by ceiling effects, in the experimental group, where the only (weak) surviving significant difference was between the first and third quadruplet in the sequential test with words, where p(1vs3, SW) = 0.04.

There were no clear trends when the items were ordered in quadruplets according to the retrieval sequence. 

We do not try to quantify these effects further, as their main feature, evident in the figure, is that they are strongly suppressed by the ceiling after mnemonic training.

The distribution across participants of the number of errors in each test is plotted in Figure 9. The distributions for control and experimental subjects are well separated, irrespective of the test, suggesting a generalized benefit of the training seminar, which is not circumscribed to only some of the tests, nor to only some of the participants.

We also analyzed, for all tests except the one with digits, how the errors were distributed between items that were among the 16, but in a different position, and other items in the pool of 48, but not among the 16. 

Across tests, retrieval errors were less frequent than position errors, ranging between 15% and 42% of total errors, once averaged for each test and subject group. We did not find any significant difference in this proportion between the tests based on sequences and those based on grids, suggesting that the simultaneous arrangement does not make it much easier to remember the relative positions, as one might expect. In detail, for experimental subjects, averaging between the W and I tests, fraction(retr_err(S)) = 0.2, fraction(retr_err(G)) = 0.3, MW-U test *p* = 0.44, while for control subjects f(retr_err(S)) = 0.25, f(retr_err(G)) = 0.36, MW-U test *p* = 0.27.

This pattern fits with the facts that (i) all words were highly familiar, whereas those particular images were (presumably) new to the subjects; and (ii) there may be some interference between remembering items in the target grid and seeing them in the pool, also a grid, which may facilitate the false memory for items not in the target grid. Figure 10 details these proportions.

## 4. Discussion

In the last few decades, WM capacity has been associated with cognitive abilities by, in particular, observing a correlation between many cognitive skills and a highly efficient WM, whereas a deficit in WM is linked to cognitive difficulties and possibly to some cognitive disorders.

As a consequence, several attempts have been made to establish the possibility of training WM, in order to benefit from its potentially enhanced performance. Unfortunately, the results have been rather controversial.

Although the exact mechanisms underlying the postulated benefits are hard to disentangle, our aim was to assess the effect of a relatively brief exposure to classic mnemonic strategies on performance in WM tasks. The strategies we shared with participants in the experimental group have been largely perfected already in antiquity, and they were certainly not designed specifically for our WM tests, but we offered them to participants in a seminar, as an option for enhancing their general memory skills, and we then tested the resulting performance in complex span tasks administered 3 h later.

Alongside the experimental group, we had a control group, whose participants attended a seminar of the same length, but which gave them only theoretical notions and no practical ready-to-use technique. In this way, we tried to control for the potential bias of using a “no-contact” group, as pointed out by von Bastian and Oberauer [15], since there was a comparable engagement and motivation for the two groups. A placebo effect, if present, was likely similar, since before taking the memory tests the two groups were only aware of participating in a memory experiment, and not that they were to be compared to the other group who had been/had not been instructed with detailed mnemonic techniques.

Notably, on average 62% of the experimental group performed at ceiling across the five tests. In the control group, we also find some subjects at ceiling, but markedly fewer, on average 20%. Interestingly, two of the control subjects, among those who performed better in that group, reported spontaneously after the tests that they had used or attempted to use classic mnemonic strategies. Thus, it appears that this form of training benefits in particular that majority of the Trieste student population who were not endowed with the education and the intellectual curiosity to refine their memory skills prior to and independently of our study.

No subject in the control group scored 100% in the word sequence test. This test in particular is also the one in which the experimental group had its poorest average score, providing evidence that this modality was the most challenging to memorize items. What is also notable in the control group responses, in fact, is that the sequential vs. grid presentation of the items had a much larger influence on average performance than the relatively minor effect of the material (words vs. digits vs. images). The benefit of the ancient strategies was distributed across all five tests for the experimental group. 

The difference between grids and sequences, particularly for the control group, could be due to the way stimuli can be visualized: as an overall, ready-to-use image for items in a grid vs. a path to follow, which has, however, to be constructed by each participant, for the sequences. Possibly, then, there is less of a need for a learnt strategy to store items presented in a grid. When asked instead to memorize a sequence, only some of the subjects, whether trained ad hoc or independently expert, succeeded in constructing a visuo-spatial path to mentally follow in order to retrieve the items. 

A further indication of the successful application of these techniques could have come from the observation of some primacy and recency trends also in the grid tests. It is well known that primacy and recency are likely to occur when participants are asked to recall items which were presented in serial order [21]. In our study they were presented, in the grid tests, together, so primacy and recency could be taken as evidence that a linear path among the items has been constructed to facilitate retrieval. We only observe weak primacy and recency trends in the control group, however, whereas in the experimental group, if potentially present, they appear suppressed by ceiling effects. In this sense, if anything, it seems as if the application of mnemonic strategies emancipates the subjects from the limitations of the simpler mechanisms involved in memorization and empowers them with a more reliable method to retrieve items.

## 5. Conclusions

Overall, our study has shown the beneficial effects of the application of memory techniques on WM tasks, evident in particular in the more challenging tests, when items were memorized in serial order. The performance of subjects who had been exposed to mnemonic techniques was largely beyond any reasonable definition of a ‘complex memory span’, in all the tasks; but also the performance of the control subjects was remarkable, in those tasks in which items were presented in a grid—suggesting that the grid presentation provides itself a simple useful spatial organization of the material to be kept in working memory. There is still ample room for improvement with appropriate strategies.

The observations above make one wonder what the ecological relevance of the complex span concept is, and whether it applies solely to the case in which no form of even rudimentary organization of the items can be utilized. For example, in remembering digits, can most subjects organize them at least into even and odd? Or phonemes, in vowel and consonants and voiced and fricatives? Or words, in so many ways? It is unclear whether models constructed around the notion of independent items can inform about any real-life situation. The same skepticism may be extended to models of working memory that derive its capacity from abstract modelling, e.g., from generic neurophysiological constraints or even from information-theoretic considerations.

The cross-task and cross-material validity demonstrated here suggests that, once assimilated, these strategies can be applied in any setting. In the future, in fact, it would be interesting to check for their effects longitudinally, and their applicability expanded in two ways: first, by measuring cognitive abilities not directly involving WM memory, i.e., their potential for far transfer [5]; second, by assessing their benefits in different subject populations. On this last point in particular, a possible development could be the involvement of populations who struggle with memory issues, such as elderly people for example. It would be very interesting indeed if, once they have acquired them, older people could use these techniques in their daily life to memorize information they may need to later recall.

In conclusion, although the debate about WM training still needs methodological clarification, we hope our study contributes to shedding light on the benefits which some mnemonic techniques can bring to WM performance. Hopefully, a short-time training can also bring long-term benefits.

## Figures and Tables

**Figure 1 entropy-22-00577-f001:**
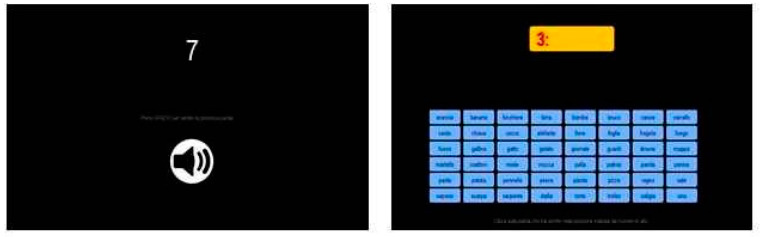
Screenshots from Test 1, Sequence of Words (SW). **Left panel**: self-paced stimulus memorization stage, **Right panel**: retrieval stage.

**Figure 2 entropy-22-00577-f002:**
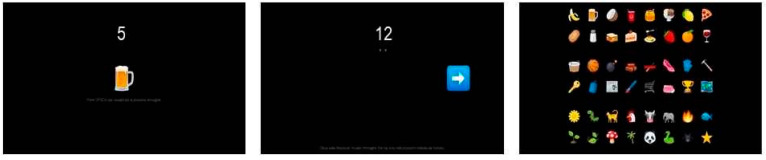
Screenshots from Test 2, Sequence of Images (SI). **Left panel**: self-paced stimulus memorization stage, item number 5 is “beer”. **Center panel**: retrieval stage, item number 12 is required. **Right panel**: retrieval stage, display of the pool of items for selection.

**Figure 3 entropy-22-00577-f003:**
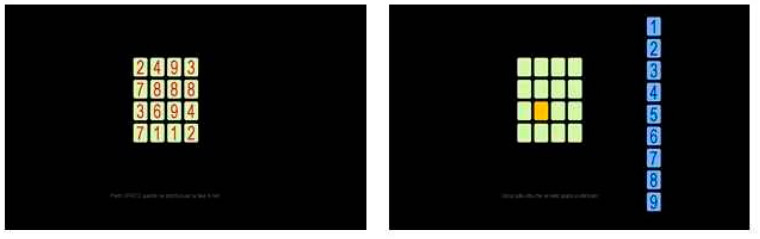
Test 3, Grid of Digits (GD). **Left panel**: memorization stage. **Right panel**: retrieval stage, the digit on the third row, second column has to be retrieved. The digit has to be chosen from the blue column on the right.

**Figure 4 entropy-22-00577-f004:**
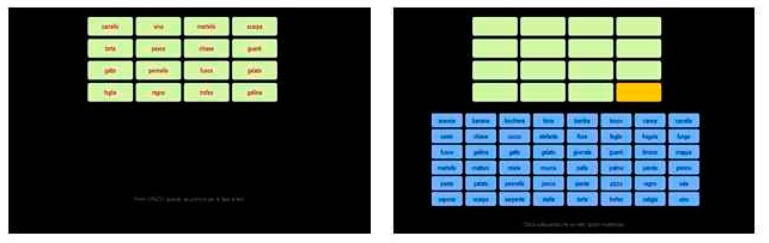
Test 4, Grid of Words (GW). **Left panel**: memorization stage. **Right panel**: retrieval stage, the word on the fourth row, fourth column has to be retrieved choosing from the pool of blue words below.

**Figure 5 entropy-22-00577-f005:**
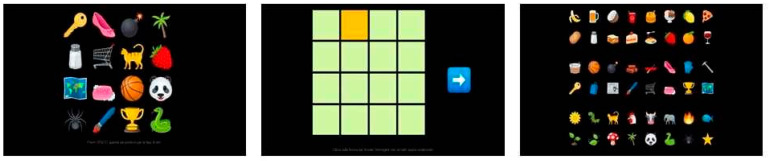
Test 5, Grid of Images (GI). **Left panel**: stimulus memorization stage. **Center panel**: retrieval stage, the image on the first row, second column has to be retrieved. **Right panel**: pool of images.

**Figure 6 entropy-22-00577-f006:**
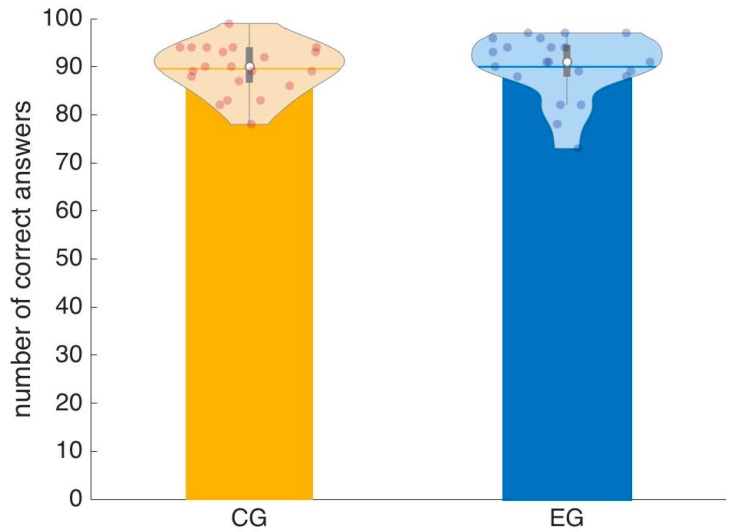
Results of the N-back experiment. There is no significant difference in the performance of control group (CG) and experimental group (EG). The violin plot uses the standard conventions: the full dots represent individual scores, the white dot represents the median and the grey thick bar represent the interquartile range.

**Figure 7 entropy-22-00577-f007:**
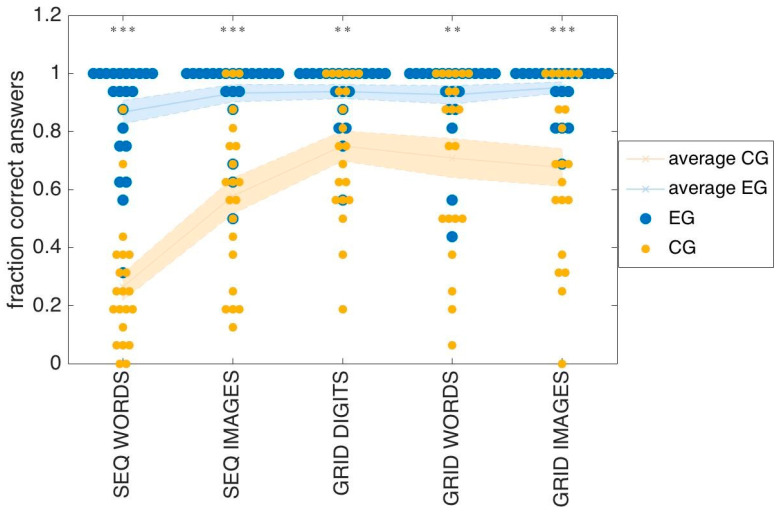
Overall performance. Each dot is the fraction of correct answers given by one subject for each of the test conditions. Yellow and blue dots represent control and experimental subjects, respectively. The average result across populations for each test condition is plotted in light yellow and light blue for the control and experimental group respectively, the shaded region represents the region mean ±SEM. The stars on top of each column represent the significance level of the difference between experimental and controls.

**Figure 8 entropy-22-00577-f008:**
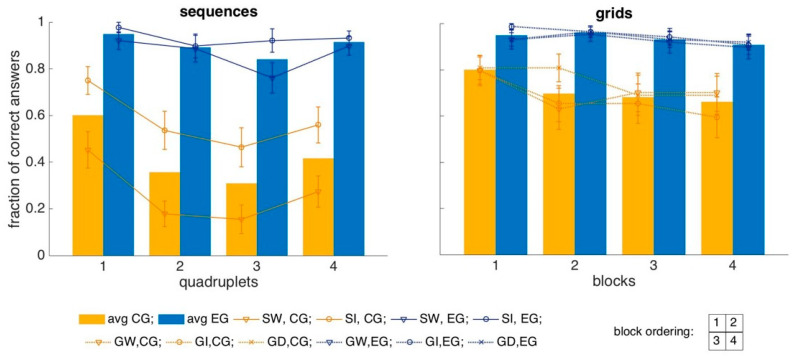
Performance in quadruplets. Yellow and blue bars represent the average across participants and across stimulus conditions of the fraction of correct answers given for each of the four quadruplets of stimuli (for the two sequence tests, they are defined by the order of presentation; for the grid tests by the quadrant on the grid). The lines report the performance separately for each of the two sequence tests, and the error bars represent the standard error of the mean across participants. “EG” = experimental group; “CG” = control group; “SW” = sequence words; “SI” = sequence images; “GD” = grid digits; “GW” = grid words; “GI” = grid images.

**Figure 9 entropy-22-00577-f009:**
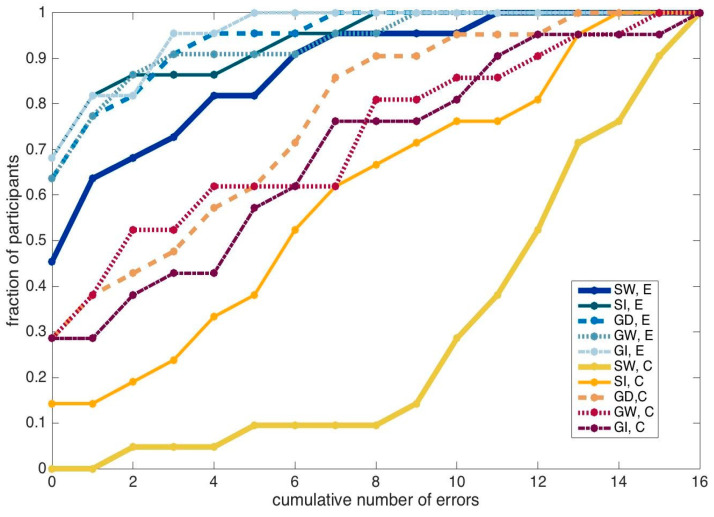
Fraction of participants that made up to a given number of errors, in the five tests. “E” = experimental group; “C” = control group; “SW” = sequence words; “SI” = sequence images; “GD” = grid digits; “GW” = grid words; “GI” = grid images.

**Figure 10 entropy-22-00577-f010:**
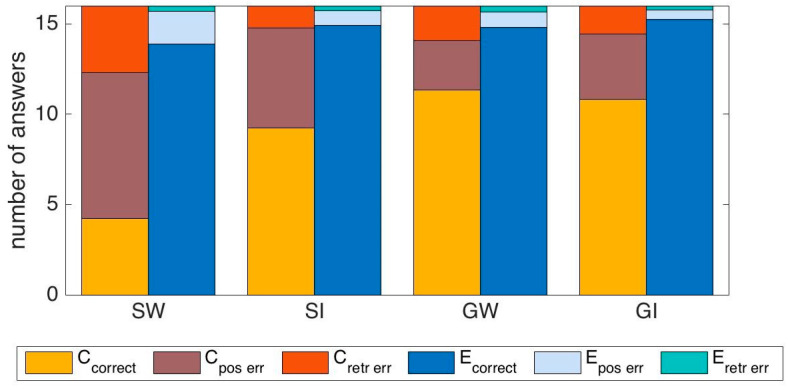
Relative frequency of correct answers and of errors, classified as ordinal position errors of items in the list of 16, and errors of retrieving items not among the 16. “E” = experimental group; “C” = control group; “SW” = sequence words; “SI” = sequence images; “GW” = grid words; “GI” = grid images.

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
