# Peer review of "Working Memory Training: Assessing the Efficiency of Mnemonic Strategies"

_entropy, 2020, doi:10.3390/e22050577_

Round 1

Reviewer 1 Report

This paper reports results of an experimental manipulation in which a control group and an experimental group are compared on a suite of working memory tasks. The experimental group is exposed to a lecturer by a skilled memorizer who teaches the participants some mnemonic techniques for improving their memory. The control group gets a control lecture.

It’s a fun idea to use an expert mnemonist in a study like this, and the results are clearly presented. In particular, I like the idea of bringing ancient cognitive techniques to bear on modern psychology questions. But I’m not sure how much we learn scientifically here.

The discussion in the intro about various WM tasks is interesting, but I found the study’s motivation a bit underwhelming: “To summarize these controversies, the road to WM training is far from being a clear path. Our aim is to explore the efficiency of some classic mnemonic strategies applied to WM tasks.”

The result is that, yes, classic mnemonic strategies improve working memory across tasks. But it is well established that these techniques work. It was even memorably documented in the book Moonwalking With Einstein in which the author (who had no mnemonic experience) becomes the US memory champ.

What more do we learn here? In a way, it seems like the experiment is a test of whether the mnemonist is an effective teacher and how quickly the teaching can happen.

If this is the point, then I think the claims in the intro should be dialed back, and it becomes less a paper about working memory and more a paper about the effectiveness of memory techniques. This is the framing that’s supported by the data. But unfortunately, that makes the whole design less about working memory and its role in cognitive psych experiments.

To be interesting from a cognitive psychology perspective, I was hoping for more explicit modeling of the different tasks in order to tell us something about how this kind of training generalizes. To that end, I think there could be more focus on the task differences. But the present study may lack the statistical power to answer that kind of question or to support more complex models.

That brings up another major point: the question of statistical power and multiple comparisons. The authors should exercise caution around the issue of multiple comparisons. In the following, there are a great many comparisons across a large number of variables: “The control group had a significantly better performance on the first and last quadruplet of sequentially presented stimuli (MW-U test, p(1&4 vs 2&3, SW&SI)= 0.0016), indicating robust primacy and recency effects. When analyzed in more detail, the effect was salient with words p(1&4 vs 2&3, SW)=0.003 but only marginal with images, p(1&4 vs 2&3, SI)=0.058, and it was largely due to differences in performance indicating primacy, in particular between the first and third quadruplets (p(1vs2, SW) = 0.009 and p(1vs3, SW)= 0.003 with words, while p(1vs2, SI)=n.s. and p(1vs3, SI)=0.01 with images; all other quadruplet comparisons were not significant).’

We should expect at least some of those comparisons to be significantly different by chance. To be sure of these kind of effects, I’d recommend spelling out hypotheses in advance, pre-registering them, or at least attempting to correct for multiple comparisons.

A couple thoughts on the figures:

-Figure 8: I’d recommend error bars on the graphs. And I’d avoid 3D bar plots which make it very hard to make the direct comparisons of interest.

-Figure 9: This could be clearer if the names were more intuitive and the labels appeared on the plot. As it is, it’s hard to figure out which line corresponds to which condition.

Finally, there are a few places where the English could be improved.

Reviewer 2 Report

Serena di Santo and her colleagues investigated the efficacy of a 2-hour training based on mnemonic strategies (vs. a control training based on a lecture on memory systems) in improving working memory (WM) in young healthy subjects. They found that WM performance, as assessed in 5 tests differing in presentation modality and retrieval demands, was better in subjects who had received the mnemonic strategy vs. control training, suggesting a training-induced improvement of WM.

This study is interesting and written well, and the findings are clear. These are my few concerns:

1 The paper puts a strong emphasis on WM, but I do not think this theoretical framing is correct. While the n-back task used to assess participants pre-training is clearly a WM task, the 5 tasks used to assess WM post-training are not. WM is about holding online a limited amount of information for a short time while operating on it (as correctly stated in the Intro). Here, participants study 16 items, which exceeds what is considered WM capacity, and additionally encoded some contextual features of the items, such as temporal and spatial position. Presentation times and the delay between study and test are not given (and should be), which makes a task analysis difficult, but in some cases presentation times were self-paced, which is also unusual for WM tasks, in which there has to be some time pressure to capture the short-term maintenance of items in WM (by definition). These tasks are not even 'complex span' tasks, as p. 3 reports, because there, too, trials involve a limited amount of information presented for a short time, although followed by an intervening complex task between trials. To me, this is an associative (long term?) memory task. Of course, this does not undermine the value of the study, which is carefully conducted, but I would adjust its theoretical framing.

2 Also, while the authors emphasize in their Introduction that a WM training should insist on WM processes (p.3), this is not the case here, because during the training participants did not complete WM tasks, but rather they were trained on visual imagery. Clearly, manipulating images requires WM (see below, point #4), but this trining is not primarily one of WM. This is a study of how promoting mnemonic strategies at encoding improves (long-term?) associative memory.

3 Please add effect sizes. You'll see they are huge compared to other studies (which you could discuss more), especially considering this is just a 2-hours training, and for the most part consisting in listening to someone rather than actually practicing WM abilities. In this respect, even if I appreciate the presence of a control training, I wonder whether the differential effect of the mnemonic and control training was due in part to the different persons delivering the training. One was a professional mnemonist who must have emphasized (memory) performance, and how to improve it, potentially conveying more motivation. Gender effects may also play a role here: the two authors delivering the trainings differed in gender, while attendees were mostly females. Could some aspects of the training be aspecific? I think this is a limitation of the study.

4 People with greater WM ability might have benefited more from the mnemonic training, because operating on mental images engages WM. In this sense, good that the groups were matched for WM performance pre-training. Was performance in the 5 post-training tasks correlated to pre-training WM performance in the experimental and (less so) in the control group? This would draw some link between WM and the mental imagery training.

5 The authors touch on generalizability of the training effect. The loci method emphasizes spatial associations. Did they expect an improvement of memory for temporal order in addition to spatial position? Why was the loci method expected to have an impact on the encoding of temporal sequences? This is important to specify because otherwise it is difficult to attribute the general effect of training across the 5 tasks to generalization or unspecific effects at point #3. A more detailed description of the contents of the mnemonic training would also help.    

6 Considering that the mnemonic training emphasized spatial encoding, were small numbers appearing early in time in the sequence remembered better than those appearing in the later positions (akin to the SNARC effect) in the mnemonic training but not (or less so) in the control group?

Round 2

Reviewer 2 Report

Satisfied with the changes. Last suggestion is to add effect sizes.

This manuscript is a resubmission of an earlier submission. The following is a list of the peer review reports and author responses from that submission.

Round 1

Reviewer 1 Report

The present work aimed at assessing the effects of mnemonic training (2 h seminar) on working memory (WM) performance in young individuals. The authors indicated that trained subjects show better WM performance in comparison to control subjects. However, the perspective of this reviewer is that this study has severe limitations in its design and execution that would question the tenability of the conclusions presented. The significance of the findings are limited and mechanism is unknown. The content of each seminar (section 2.4) and its logic are unclear. In addition, statistical information is lacking. Until the statistics are properly reported, this manuscript cannot be properly reviewed.

Reviewer 2 Report

Table of Contents
─────────────────

1. Overview
2. Major concerns
.. 1. Statistical analysis
.. 2. Scope of the paper and match with the journal
3. Minor concerns
.. 1. Data availability
.. 2. Subject-level information missing from figure 6
.. 3. Clarity, references and typos

1 Overview
══════════

In this paper, the authors present an experimental study on the
effectiveness of working memory training, based on classic techniques
such as the "memory palace". The study is designed to reduce at a
minimum some of the confounds identified by previous work in the
literature. The outcome of the experiment, the authors claim, strongly
supports the hypothesis that working memory can be improved by
training, and that methods such as those employed are very effective
in doing so. The paper concludes by briefly mentioning some
interesting extensions of the present study.

Overall, I have found the paper to be well written, the problem
clearly stated, the experiment well designed and the results very
interesting. However, I have two concerns to raise (besides a few
minor suggestions) before I can recommend this work for publication.

2 Major concerns
════════════════

2.1 Statistical analysis
───────────────────────────────────────

My first major concern is methodological in nature, and I believe it
should be easily addressable by the authors in a revised
submission. Briefly, the data presented looks good in the plots, but
it should be supported by at least a minimum degree of statistical
analysis showing quantitatively that the data supports the claimed
effects. More precisely, summary quantities in figures 7, 8 and 9 are
all missing uncertainty estimates (error bars), and no indication is
given about the statistical significance of the differences between
treatment and control group. The authors should devise appropriate
statistical tests to back up their claims. Some examples:
• on line 256, the difference between the two groups is claimed to be
"clear from the figure": this type of statement should be backed up
statistically throughout the paper, by some quantitative comparison
to the difference that may be expected by chance (or by some other
appropriate statistical device).
• line 291-292: "the distributions.. are well separated": same as
above, and to all other statements of this sort.
• in the caption to Figure 6, it is claimed that "there is no
difference in the performance" of the two groups, but the figure
only reports a measure of central tendency and dispersion for each
group (I assume these are mean and standard error of the mean,
though no description is given). Needless to say, one can give
infinitely many examples of distinct distributions with given mean
and variance, so the authors should specify the measure according to
which the two groups show no difference (see also my other comment
below on Figure 6, in "minor concerns").
More generally, all statements of the same type as the above should be
appropriately backed up by quantitative analyses.

2.2 Scope of the paper and match with the journal
─────────────────────────────────────────────────

My second main issue concerns the thematic match with the topics
covered by the journal. In particular, the present paper, though valid
and interesting per se, lacks any reference to information theory
(apart from an opaque, cursory and dismissive mention in the
discussion), or any other theoretical framework that may be of
interest to the journal's readership. For reference, the "aims and
scope" page of entropy's website states that "Relevant submissions
ought to focus on one of the following: develop the theory behind
entropy or information theory; provide new insights into entropy or
information-theoretic concepts; demonstrate a novel use of entropy or
information-theoretic concepts in an application; obtain new results
using concepts of entropy or information theory". If this were a
regular submission, this mismatch would constitute sufficient grounds
for recommending rejection. However, since this work has been
submitted to a special issue on the topic of working memory, I will
defer to the judgement of the issue editors on this point.

3 Minor concerns
════════════════

3.1 Data availability
─────────────────────

In the interest of openness and reproducibility (and as required by
the journal's guidelines), the authors should make their raw data
available online, possibly in a non-proprietary format. I suggest that
this is done via a purpose-built repository such as Zenodo
(zenodo.org) or Data Dryad (datadryad.org), or by submitting the data
as a (set of) supplementary file(s).

3.2 Subject-level information missing from figure 6
───────────────────────────────────────────────────

By only giving group level means and standard errors (I assume the
error bars represent standard error of the mean - this is not reported
anywhere), Figure 6 is not very informative of the actual data it
represents, especially given what should be possible due to the rather
small number of subjects. For example, the performance of each
individual could be shown instead of (or in addition to) the
group-level mean, or some more information-dense plot type could be
used (e.g. violin plot). Note that this is not just an aesthetic
consideration: indeed, the absence of subject-level data in this plot
makes it hard to compare it to Figure 7. The mean performance of both
groups is at 90%, leading the reader to think that this may be due to
both groups being essentially at ceiling performance. But this is not
the case in Figure 7: the homogeneity between the two groups is being
assessed in Figure 6 with a task that is significantly easier than the
ones performed later in the experiment, making this control weaker
than what was perhaps intended by the authors. Having more information
on the shape of the distribution of performance across subjects in the
two groups in Figure 6 may make it easier for the reader to convince
themselves that no identifiable patterns are present that could
explain part of the differences in Figure 7.

3.3 Clarity, references and typos
─────────────────────────────────

• Line 41: "Most scholars agree…". Please avoid weasel words. If the
authors want to claim that broad consensus exists within the
community around a certain idea, they should provide specific
references to support their claim.
• Lines 76 and 94: As far as I understand this section of the
introduction (spanning more or less three paragraphs) follows mostly
the arguments raised by von Bastian and Oberaurer. This is fine, but
please signpost this better as it may be confusing to the
reader. The initial, explicit reference to von Bastian and Oberaurer
is done inside the paragraph starting on line 68 ("According to von
Bastian and Oberaurer [13]…"), and then the discussion of this paper
is continued throughout the following two paragraphs, with the
references becoming progressively more implicit (line 76: "In their
review [13], the authors…"; line 94: "…the authors…"). But these are
all independent paragraphs (i.e., they are not organized in a
section/subsection hierarchy), so in principle they should be
logically self contained, or their dependencies should be made
explicit. In other words, when reading "the authors" on line 94 (but
also on line 76), the reader may be surprised or confused that the
discussion of von Bastian and Oberaurer did not terminate within the
paragraph where it was initiated. I believe just repeating "von
Bastian and Oberaurer [13]" each time (or something to that effect)
would be much clearer.
• Line 214: lighted → lit
• Line 243: 89%±5%, 90%±6%. Are these mean±s.e.m.? please report the
meaning of the summary quantities and of their dispersion measures.
• Figure 8: unless I'm reading the figure wrong, I am confused by the
reference to a broken line in the legend. There seems to be no
broken line in the figure.
• Figure 8: please place labels (at least what is blue and what is
yellow) *on the figure itself*, rather than forcing the reader to go
back and forth from the legend.
• line 342: "the benefit [was]… more salient in those further to the
ceiling in the control group". It is not clear what the authors mean
with this statement. Of course there is more room for improvement in
the tasks where the baseline is lower than in those where it is
already close to the ceiling, so the fact that the improvement is
indeed larger in these cases is hardly surprising. Is this intended
to mean that the benefit was *even larger* than what one could have
expected given the difference in room for improvement, according to
some model of how this benefit would play out? please clarify, and
provide adequate quantitative support if necessary (i.e. in case my
latter interpretation corresponds to the authors' intention).